# DIFAIR — Towards learning differenciated and interpretable representations

## Abstract

Neural network classifiers are generally trained to differentiate between the same classes during training and testing. In order to prevent incorrect predictions, when an input image contains a class that was not part of the training set, it should be detected. The process of detection of "unknown" classes is called *Open-Set Recogniton* (OSR). Given that a neural network extracts a representation (a feature vector) describing an image, its capacity to detect the presence of a class in an image, through the recognition of specific features, should also imply the ability to detect the absence of a "known" class, through the absence of those features in the representation. In this article, we present DIFAIR, a novel approach introducing the key characteristics that a feature representation should exhibit to ensure: (i) class separability, through predefined class positions in the representation space; and (ii) interpretability by associating each dimension of the representation with a class. We present a loss function to optimize a model, in a supervised way, in order to produce the proposed representation. Our approach assumes that unknown classes should share only a limited number of features with known classes and therefore we evaluate its performance in OSR. Finally, we visually inspect learned representations to identify the flaws of our loss function and present directions for future improvement.

## 1 Introduction

Image classification is a crucial task in vision recognition, aimed at characterizing images and categorizing them into specific classes (Wang & Su, 2019). Recent advancements in this field have been propelled by the emergence of deep neural networks. Convolutional neural networks, including VGG (Simonyan & Zisserman, 2014) and ResNets (He et al., 2016), have been used extensively and inspired many variants. Moreover, the adaptation of the transformer architecture (Vaswani et al., 2017) to image classification by (Dosovitskiy et al., 2020) has achieved state-of-the-art performance. These models are trained using the cross-entropy loss function and employ the softmax function in the final layer to estimate class probabilities. This setup forces the network to make predictions even when presented with images featuring classes absent from the training data, referred to as "unknown classes" in this paper. In this scenario, models are assumed to predict on the *same classes* during training and testing phases, a concept termed *closed-set classification* (Scheirer et al., 2013).

However, using a model designed for closed-set scenarios is inadequate for real-world applications, where the model operates in dynamic environments and encounters classes that might change. Furthermore, real-world applications often involve a significantly larger number of classes than those typically used for model training. It is unreasonable to expect a dataset that encompasses every conceivable object, both existing and future ones: hence the need for robust deep neural networks capable of handling unknown data. To address this issue, Scheirer et al. (2013) introduced the notion of *Open-Set Recognition* (OSR), where unknown classes are presented to the model during the test phase with the dual objective of detecting them while still being able to classify known classes.

As discussed by Dieterich & Guyer (2022), we hypothesize that if a model is capable of recognizing the presence of a class in an image, it should somehow be able to discern its absence. More precisely, if a model can identify features specific to a particular class, then when that class is not present in the image, the features that the model has learned for it should remain inactive, approaching values close to 0. Empirical experiments have supported this hypothesis, demonstrating that the norm of the

feature vector when dealing with unknown classes is often lower than that for known classes (Chen et al., 2021; Vaze et al., 2022). It is also possible that new classes share certain features with known classes. In such cases, the model may misclassify instances, leaving us without a clear interpretation of its decision. When examining the feature vectors learned by most neural networks, it becomes evident that we cannot readily discern whether an image belongs to known classes, primarily because we lack information about which features describe a particular class. Furthermore, the challenge is compounded by the fact that features can be shared by multiple classes due to the distributed representations inherent in neural networks (Hinton, 1984), making the interpretation even more challenging.

From these observations, two objectives emerged: (i) enabling the model to extract features that describe a specific class; (ii) ensuring that features absent in the image remain "disabled", by approaching values close to 0. Thus, the main goal of our work is to attain a *more interpretable representation*, that has class-associated features and is free from distributed representations. We called our approach DIFAIR, standing for *DIFerentiated And Interpretable Representations*.

To achieve these objectives, we aim to control the representation space by defining an association between specific dimensions and classes. Each dimension associated with a class should represent *different features* extracted for that class. This is valuable for OSR, as we hypothesize that an unknown image will activate features from different classes or no features, allowing its detection. Figure 1 schematizes our desired organizational structure for these representations.

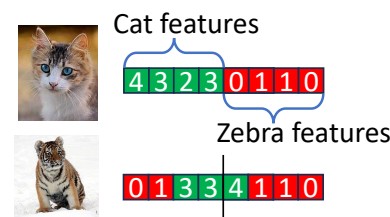

Figure 1: Example of our intended feature activation using DIFAIR. Extracted features are class-associated and should be activated only when present in the image. For this reason, a model trained on cats and zebras may detect common features when presented with an image of a tiger.

Existing approaches in the domain of OSR have proposed some methods to regroup representations of known classes in space to be more robust to unknown inputs. Whether through optimization around predefined class centers (Miller et al., 2021) or learned prototypes (Chen et al., 2021; Cevikalp et al., 2023), to the best of our knowledge none of these approaches have specifically been optimized for more interpretable representations.

In this paper, our contributions on DIFAIR encompass three key aspects: (i) we introduce a loss function designed to optimize the network's learned representation, aligning it with the constraints specified above; (ii) we use OSR tasks as a means to assess the quality of the learned representation, since our loss function organizes how known instances are distributed in the latent space, we anticipate favorable results in OSR; (iii) we visualize the learned representations and illustrate how they offer insights into the model's behavior compared to representations derived from standard models.

## 2 RELATED WORK

### 2.1 OPEN-SET RECOGNITION

Scheirer et al. (2013) introduced the concept of open-set recognition (OSR), a more realistic classification scenario than traditional closed-set recognition. OSR expects classifiers to not only classify instances belonging to known classes, but also detect unknown classes at test time. This stands in contrast to closed-set recognition where a model is evaluated only on classes that were seen during training. Bendale & Boult (2016) were the pioneers in applying OSR within the domain of deep learning, introducing the OpenMax approach, which involves measuring the distance between a new instance's point and the mean activation points of known classes in the logits space. Using Extreme Value Theory on distances, they calibrate the network prediction in order to predict a probability of belonging to an unknown class. Other approaches have exploited Generative Adversarial Networks (GANs) to generate data representing unknown classes, including works by Ge et al. (2017); Neal et al. (2018); Chen et al. (2021); Moon et al. (2022). These approaches train the network to make specific predictions when confronted with unknown instances. Additionally, real data from external datasets can be employed as examples of unknown classes during the training phase, as in (Cevikalp et al., 2023). Other OSR strategies involve learning prototype points in the latent space to

represent either known or unknown classes, as explored by Liu et al. (2022) and Chen et al. (2021), respectively. The distance to these prototypes is subsequently used to detect unknown instances.

To the best of our knowledge, current state-of-the-art methods designed for OSR include ARPL+CS (Adversarial Reciprocal Points Learning) (Chen et al., 2021), OpenHybrid (Zhang et al., 2020) and DCHS (Deep Compact Hyperspheres) (Cevikalp et al., 2023). Vaze et al. (2022) found a correlation between closed-set accuracy and open-set performance, assessed using the area under the receiver operating characteristic (AUROC) for the unknown detection task. Their study reveals that a baseline network trained with cross-entropy loss can match or surpass the performance of state-of-the-art methods. This achievement requires enhanced training with data augmentation, an increased number of training epochs, and a specific learning rate schedule. We note that ARPL+CS and DCHS incorporate supplementary data as unknown instances in training, while Vaze et al. (2022)'s baseline exclusively trains on known classes.

## 2.2 CLASS ANCHOR CLUSTERING

We focus on a specific OSR method called Class Anchor Clustering (CAC) from Miller et al. (2021), that presents ideas close to ours, such as organizing the representation space using anchors. CAC anchors class representations in the final layer of the neural network, in the logit space. Essentially, for each class, anchors are represented as one-hot vectors, scaled by a parameter $\alpha$. The objective of the network optimization is to produce logits that are in proximity to these anchors in terms of Euclidean distance. The loss function used in this approach comprises two components: (i) the anchor loss, a component used to minimize the distance between an instance's representation and its corresponding anchor and (ii), a modified *Triplet Loss* (Sohn, 2016), a component used to maximize the difference between the instance's distance to its true anchor and its distance to all other anchors.

During the testing phase, the distances $d$ between predicted logits and each class anchor is measured, and subsequently used to compute a rejection score $\gamma = d \circ (1 - \mathrm{softmin}(d))$ to express the degree of uncertainty regarding whether the input belongs to a known class. The $\mathrm{softmin}(x)$ function, equivalent to $\mathrm{softmax}(-x)$, is used to assign the highest value to the lowest distance. Therefore, for an input to be recognized as *known*, it must exhibit both a low distance to an anchor and a high $\mathrm{softmin}$ score. In conclusion, if $\min(\gamma)$ exceeds a predetermined threshold, the prediction is rejected as *unknown*; otherwise, the prediction is accepted for the class with the lowest rejection score. According to the authors, employing such anchored representations lacks semantic meaning, as anchors are uniformly distributed in the representation space. They suggest that incorporating more semantic features could enhance OSR performance. In line with this observation, our approach, DIFAIR, aims to introduce additional semantic meaning and interpretability to the learned representations.

## 3 METHODS

### 3.1 OSR FORMALIZATION AND NOTATIONS

A neural network classifier is defined by $\mathcal{C}(\boldsymbol{X}) = c \circ f(\boldsymbol{X})$, where $c$ is the classification head (generally a fully connected network), $f$ is the feature extractor (generally convolutional layers) and $\boldsymbol{X}$ is an input image. We note $\boldsymbol{z} = f(\boldsymbol{X})$ the *representation* learned by the feature extractor and $\hat{\boldsymbol{y}} = c(\boldsymbol{z})$ the *logits* output of the classifier, i.e. no activation function is applied on $\hat{\boldsymbol{y}}$.

In this work, we follow the formalization of OSR proposed by Vaze et al. (2022) and compare it to closed-set recognition. Let $\mathbb{X}$ be the input space (images in our case), and $\mathbb{C}$ is the set of "known" classes. In *closed-set recognition*, a classifier is trained on $\mathbb{D}_{\text{train}} = \{(\boldsymbol{X}_i, y_i)\}_{i=1,\dots,n} \subset \mathbb{X} \times \mathbb{C}$ and evaluated on $\mathbb{D}_{\text{test-closed}} = \{(\boldsymbol{X}_i, y_i)\}_{i=1,\dots,m} \subset \mathbb{X} \times \mathbb{C}$, where all test inputs belong to known classes. The closed-set classifier predicts a score for each known class reflecting the input's association with that particular class. This score is often a probability distribution $p(y|\boldsymbol{X})$. OSR considers a more realistic scenario where inputs from any classes could be given to the classifier. Therefore, in addition to doing classification, OSR aims to detect inputs belonging to "unknown" classes. In this situation, the classifier is trained on $\mathbb{D}_{\text{train}}$ and is evaluated on $\mathbb{D}_{\text{test-open}} = \{(\boldsymbol{X}_i, y_i)\}_{i=1,\dots,m'} \subset \mathbb{X} \times (\mathbb{C} \cup \mathbb{U})$ with $\mathbb{U}$ the infinite set of all unknown classes. In OSR, supplementary data containing "known unknowns" (data identified as unknown) can be used to train the model to reject predictions on unknown data (Cevikalp et al., 2023).

In the open-set setting, given an input $\boldsymbol{X}$, the classifier is expected to return a score for each known class *and* a score $S(y \in \mathbb{C}|\boldsymbol{X})$ used to detect whether or not $\boldsymbol{X}$ contains any of the known classes. An example of such a score is $S(y \in \mathbb{C}|\boldsymbol{X}) = \max \mathrm{softmax}(\hat{\boldsymbol{y}})$, which is the highest probability output of a network trained with cross-entropy loss. Instances scoring above a given threshold will be considered as *known* by the network and instances scoring below as *unknown*. Varying the threshold on the score allows to compute AUROC for the task of unknown detection.

### 3.2 DIFAIR: DIFFERENTIATED AND INTERPRETABLE REPRESENTATIONS

*DIFferentiated And Interpretable Representations* (DIFAIR) aims to control the representation space $\boldsymbol{z}$ to ensure that instances of the same class activate common features while avoiding features from other classes, thus enhancing interpretability and learning differentiated class representations.

**Methodology.** As in CAC (Miller et al., 2021), the DIFAIR approach initially defines class anchors for each class as *fixed points* in space around which the network will learn to represent class instances. However, there are key distinctions that set DIFAIR apart: (i) unlike CAC, which anchors in the logit space $\hat{\boldsymbol{y}}$, DIFAIR anchors in the *feature space* $\boldsymbol{z}$; (ii) anchoring in the feature space allows to allocate multiple dimensions per class whereas CAC used only one dimension per class in the logits space. Having more than one dimension per class is advantageous as each dimension can express the presence or absence of *one* feature from this class in the image. We hypothesize that the allocation of each dimension to a specific class can yield more meaningful features that provide insights into predictions. Instead of having a fully connected layer exploiting the representation to make a prediction, we aim to force the features to be representative of a class.

In the proposed approach, the anchors are characterized by two hyperparameters: the number of dimensions, $\mathcal{N}$, allocated per class, and $\alpha$ the desired activation value on allocated dimensions. For example, in a classification problem with three classes and $\mathcal{N} = 2$, class anchors would be at the coordinates in Eq. 1, in which non null dimensions are considered associated with the respective class, because the objective will be to activate these dimensions when presented with the class.

$$\mathcal{A}^1 = (\alpha, \alpha, 0, 0, 0, 0) \quad \mathcal{A}^2 = (0, 0, \alpha, \alpha, 0, 0) \quad \mathcal{A}^3 = (0, 0, 0, 0, \alpha, \alpha) \tag{1}$$

Currently, we allocate the same number of features to all anchors and they have the same norm. With these representations, all anchors are equidistant, as in CAC. However, Miller et al. (2021) note that in such a situation, extracted features lack semantic meaning. To allow our representations to bear semantic meaning, we allocate a hypersphere around each anchor. This hypersphere defines a space in which instances of a class can be represented. The radius of the hypersphere, denoted $r$, is an additional hyperparameter. More details on the semantics brought by the usage of a hypersphere are provided in Section 3.3. Figure 2 illustrates this idea in comparison to the standard learning setting.

To make a prediction using this representation, the Euclidean distance between the representation of a given instance and every anchor $\boldsymbol{d} = \mathrm{Eucl}(\boldsymbol{z}, \mathcal{A}) = (\|\boldsymbol{z}, \mathcal{A}^0\|_2, \ldots, \|\boldsymbol{z}, \mathcal{A}^{\#\mathbb{C}}\|_2)$ is measured. The predicted class $y$ is determined by $y = \arg\min \boldsymbol{d}$. The measured distances are used to compute the OSR score $S(y \in \mathbb{C}|\boldsymbol{X}) = \min \boldsymbol{d}$, meaning that if an instance is not within a determined radius of an anchor, the prediction is rejected. Unlike CAC, anchors are not adjusted at the end of learning, since it would make it harder to interpret features to class association.

DIFAIR can fit any convolutional architecture, as long as the classification head $c$ is removed and a convolutional layer which will output $\mathcal{N} \times$ number_of_classes filters is added, assuming that each of these filters detects a specific feature in the image (Géron, 2019). After the added convolution layer, global average pooling (Lin et al., 2013) is used to obtain the vector of features $\boldsymbol{z}$.

**Learning with DIFAIR.** To optimize our model using this method, we use as a loss function the Euclidean distance between the predicted representation $\boldsymbol{z}$ and the true class anchor $\mathcal{A}^c$, thresholded by $r$ (see Eq. 2). This ensures that the loss value is 0 when an instance is within the hypersphere, and the value becomes the distance to the true class anchor otherwise. The fact that the loss is set to 0 when the distance is below $r$ allows the network to represent the instance anywhere within the hypersphere: there are *no constraints* on the representation as long as the instance remains within the anchor's radius. Representing data in hyperspheres avoids the optimization of an invariant representation for each class, which is the case in CAC. This enables the network to find representative

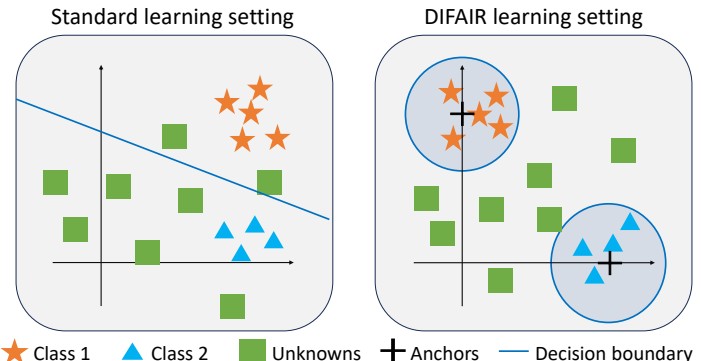

Figure 2: **Comparison of a standard representation space with the representation space learned by DIFAIR.** (Left) Standard representation space learned with cross-entropy loss. It lacks a guarantee that dimensions will represent independent features. We note that similar classes are in close proximity to each other, and unknown data points exist on both sides of the decision boundary. (Right) Representation space learned by DIFAIR. Known instances are clustered around defined anchors within a hypersphere. Unknown instances sharing features of both classes are represented outside of hyperspheres. Each dimension is directly responsible for the presence or absence of a feature: if values on these dimensions are too low, instances will not be in any hypersphere.

features for each class, and embed semantic information in the representation (section 3.3).

$$\mathcal{L}_d = \max\left(\sqrt{\sum_i (z_i - \mathcal{A}_i^c)^2} - r, 0\right) \tag{2}$$

During training, we use only $\mathbb{D}_{\text{train}}$ and no "known unknown" data. We justify this choice to verify the hypothesis that a classifier which extracts semantically significant features from its training data can also detect the absence of such features at test time. Learning with cross-entropy loss does not force features to be completely at 0 when a class is absent from the input image, it is only necessary for the features of the true class to be more activated than other features. Learning using Euclidean distance as a loss function is a bit different from cross-entropy learning. If there was no hypersphere, the features from other classes should all be 0 for a representation to be close to its anchor. However, having instances represented within a hypersphere allows to have some values activated on other dimensions. Depending on the radius of the hypersphere, our approach is more or less tolerant to the activation of other class features. This assumption is discussed in Appendix B.

### 3.3 LEARNING HYPOTHESIS

In this section, we emphasize some assumptions and hypotheses that underlie this work, explaining why DIFAIR can be applied to OSR.

**Interpretability.** Our main idea is that if a feature is present in a known image, a neural network representation $z$ should have an associated dimension in its representation that outputs a high value. If the feature is absent, this value should be close to 0. Therefore, using class anchors with $\mathcal{N}$ dimensions associated with each class facilitates the creation of more interpretable representations. This approach ensures that, for an input image, all detected features can be associated with a class. Since we want features to be activated only for one class (with some tolerance thanks to the hypersphere), our method should extract independent features for each class. For example, in an unconstrained neural network, one activated feature could be representative of the dog class, while the same feature (possibly with a different activation level) could also be representative of the cat class. We would like to have this feature learned on two different dimensions, to know explicitly that it is common to two classes. We note that while our preference is to have only one feature recognized per dimension, we do not explicitly optimize for this. Thus, the network can technically learn to detect a group of features in an image and express them on one dimension. Therefore, a class would

be described by multiple groups of features, each associated with a dimension. However, one group of features should be associated with only one class.

**Semantically meaningful representations.** We consider that semantic similarity in a representation is equivalent to having proximity between representations. In a setup like CAC, where class anchors are equidistant and instances are optimized to be clustered *on* the anchors, if the global minima for the optimization problem is found, all instances from a class are represented at the anchor in the latent space. Therefore, there is no proximity between similar classes, thus no semantics in the representation. On the other hand, if hyperspheres are allocated around anchors, in the same optimization setting, all points are within the hypersphere but not necessarily grouped only at the anchor. From there, it is possible (but not certain) for a point representing a cat instance to be located in the hypersphere on the dog's anchor side and there would therefore be a proximity with dog class for this instance. In order to have this proximity (and a semantic meaning), some features of the dog class should be activated in the cat instance representation. The activation of features from other classes is possible thanks to the usage of the Euclidean distance with a threshold (defining a hypersphere). For those reasons, we conjecture that the DIFAIR-learned features can retain a semantic meaning, enhancing the interpretability and the richness of the representations.

**Open-Set Recognition.** We hypothesize that our method clusters data around well-separated anchors in space, and extracts independent features representing classes, which are activated when features are actually present in the images. From there, an image containing an unknown class should not activate *all* features from one known class, otherwise the class would not be unknown. Instead, an unknown class might activate features of multiple known classes, or none at all if the image lacks those specific features. Based on these considerations, we hypothesize that the Euclidean distance is a suitable score for our problem. For a new instance, the more features from one class are activated, the closer the instance will be to this class anchor. If features from different classes are activated, the instance representation might not be close to any anchor. In this situation, it will be detected as unknown thanks to a chosen threshold on the distance. For these reasons, we believe that our method holds the potential to yield good results in OSR.

## 4 OPEN-SET RESULTS

With DIFAIR, we aim to learn well-separated representations in space as well as class-representative features on each dimension. To assess our approach in OSR, we adhere to Neal et al. (2018) standard evaluation protocol, using their proposed network architecture. It is a compact VGG architecture (Simonyan & Zisserman, 2014), referred to as "VGG32" (refer to Appendix A.1 for details).

**Methods.** We use as a baseline a cross-entropy trained neural network, denoted as $\mathcal{C}_{CE}(\boldsymbol{X}) = c(\boldsymbol{z}_{CE})$. From this baseline, we extract multiple OSR scores: the Maximum Softmax Probability (**MSP**) with $S(y|\boldsymbol{X}) = \max \text{softmax}(\hat{\boldsymbol{y}})$ and the Maximum Logit Score (**MLS**) proposed by Vaze et al. (2022) which uses $S(y|\boldsymbol{X}) = \max \hat{\boldsymbol{y}}$. To better compare the DIFAIR representations to the baseline, we extract the representation $\boldsymbol{z}_{CE}$ from $\mathcal{C}_{CE}$, and report results with $S(y|\boldsymbol{X}) = \min \text{Eucl}(\boldsymbol{z}_{CE}, \boldsymbol{C}_z)$, where $\boldsymbol{C}_z$ is a matrix containing the average representation for each known class. We refer to this experiment as *Cross-Entropy Loss Representation* (**CELR**). We generated new results for Class Anchor Clustering (**CAC**) (Miller et al., 2021) to ensure a fair comparison with our proposed approach because the original results were generated on different random splits, which can influence performance. For CAC, we used the OSR score reported in their article (described in Section 2.2). Additionally, state-of-the-art results from other papers are reported: (1) Compact Hyperspheres (**DCHS**) (Cevikalp et al., 2023). This method, which shares some similarities with DIFAIR, learns the position of class centers in the representation space. It uses supplementary data from the 80 Million Tiny Images dataset (Torralba et al., 2008) to represent unknown classes during the training phase, potentially exposing the model to some unknown classes that will have to be detected; (2) (**ARPL+CS**)+, proposed by Chen et al. (2021) and improved by Vaze et al. (2022), uses learned prototypes to represent reciprocal points from which class instances must be distant. It uses Confusing Samples (CS) generated by a GAN as supplementary unknown data. We also test (for CAC and DIFAIR) the open-set performances using Maximum Output Score (**MOS**), a generalized version of the Maximum Logit Score from Vaze et al. (2022) that is applied either on

a logit vector or a representation, instead of using the distance to the closest center. In our reporting below, methods using MOS are indicated by a '†' next to the method name.

**Datasets.** Neal et al. (2018) introduced a benchmark to evaluate OSR performances. The benchmark aims to use a standard closed-set dataset, split the classes into 2 random groups, and use one group as known classes for training, while the other group contains unknown classes to be detected when testing. Results are then averaged on 5 random splits. We use the same splits as Vaze et al. (2022) to obtain comparable results. MNIST (LeCun et al., 2010) and SVHN (Netzer et al., 2011) contain digits from two different domains: handwritten and from street view house numbers, respectively. CIFAR10 (Krizhevsky et al., 2009) contains 10 classes of animals and vehicles. Neal et al. (2018) proposed a setting where the 4 vehicle classes from CIFAR10 are used as known classes and $N$ random classes from CIFAR100 (Krizhevsky et al., 2009) are used as unknown classes. $N$ can be either 10 or 50 so the experiments are referred to as CIFAR+$N$. It is important to note that classes from CIFAR10 and CIFAR100 are non-overlapping, ensuring that unknown classes are never seen during training. Lastly, TinyImageNet (Le & Yang, 2015), a dataset of 200 classes subsampled from ImageNet (Russakovsky et al., 2015) is used in a setting where 20 random classes are known, while the remaining 180 are considered unknown.

**Experiments.** We report results obtained by Cevikalp et al. (2023) and Vaze et al. (2022) for both DCHS and (ARPL+CS)+. For our experiments and CAC re-training, we follow the improved training protocol proposed by Vaze et al. (2022), which involves training for a greater number of epochs (600), using RandAugment data augmentation (Cubuk et al., 2020), and a learning rate scheduler. For DIFAIR, we used $\mathcal{N} = 5$, $\alpha = 10$ and $r = 0.4 \times \sqrt{2\mathcal{N}\alpha^2}$. This configuration allocates 40% of the space between two anchors to each hypersphere, leaving 20% of the space between anchors for unknown data. Further details on these experiments and the hyperparameter search for DIFAIR can be found respectively in Appendix A.2 and C.

**Results.** OSR results on the Neal et al. (2018) benchmark are reported in Table 1. We encountered challenges in reproducing the same baseline results as those reported by Vaze et al. (2022) with the Maximum Logit Score (MLS), so we also reported their own results for comparison. Both CAC and DIFAIR exhibit lower AUROC scores compared to the MLS baseline, even when using the Maximum Output Score (MOS). MOS is more adapted to measure OSR capabilities of DIFAIR, exhibiting a significant improvement in AUROC compared to using the distance to anchors as a score. When comparing DIFAIR$^†$ to CAC, reported results are quite close. Additionally, using cross-entropy learned representations (CELR) to compute the OSR score yields results that are notably lower than the other methods, with the exception of the results obtained on the MNIST dataset.

Table 1: **OSR results on Neal et al. (2018) benchmark**. AUROC score for the task of detecting unknown classes, averaged over five known/unknown class splits. We report closed-set accuracies for the trained methods in Appendix D. A '†' next to the method name indicates that results were obtained using Maximum Output Score.

| Method | MNIST | SVHN | CIFAR10 | CIFAR+10 | CIFAR+50 | TinyImageNet |
|---|---|---|---|---|---|---|
| MSP | 99.4 | 95.6 | 89.0 | 90.5 | 89.8 | 66.3 |
| MLS$^†$ (re-trained) | **99.6** | 96.9 | 92.0 | 91.8 | 92.1 | 67.9 |
| MLS$^†$ (Vaze et al., 2022) | 99.3 | **97.1** | 93.6 | 97.9 | 96.5 | 83.0 |
| CELR | 99.2 | 87.1 | 57.1 | 67.2 | 69.3 | 54.3 |
| CAC (re-trained) | 98.9 | 96.6 | 86.1 | 86.5 | 87.0 | 58.2 |
| CAC$^†$ (re-trained) | 97.4 | 96.9 | 86.0 | 83.4 | 84.9 | 69.0 |
| DCHS (Cevikalp et al., 2023) | 96.6 | 94.5 | **94.7** | **99.2** | **98.5** | **83.8** |
| (ARPL+CS)+ (Vaze et al., 2022) | 99.2 | 96.8 | 93.9 | 98.1 | 96.7 | 82.5 |
| DIFAIR | 98.9 | 93.5 | 77.3 | 77.4 | 77.4 | 65.2 |
| DIFAIR$^†$ | 99.5 | 94.8 | 86.5 | 86.8 | 86.1 | 69.3 |

**Discussion.** Results using CELR show that cross-entropy representations are not optimized for OSR relying on distances. This means that cross-entropy does not create one tight cluster per class

in its representation. On the other hand, the classifier $c$ is able to extract useful information from the representation space, since MLS gives results close to the state-of-the-art.

DCHS and ARPL+CS try to regroup *all* unknown classes at one point in the representation space. While this approach leads to excellent OSR performance, it implies that there is no semantic information to be extracted from data detected as *unknown*. Whereas in our approach, features can be activated when facing unknown data since there are no constraints on unknown data during training. The aim of our approach is that if a *known* feature is present in an image, then the corresponding dimension should be activated in the representation, even for unknown images. It seems that, due to the flexibility allowed in feature activation for unknown data, using distances as an OSR score might not be the most suitable approach to detect unknown classes, as evidenced by the results of DIFAIR compared to DIFAIR[†]. The results using MOS show that, for unknown classes, features tend to be activated with a lower value than for known classes. To investigate further the reasons why DIFAIR results are lower, we visualize learned representations in Section 5.

Comparing DIFAIR to CAC, using distance to anchors score, we note that adding dimensions and allocating hyperspheres around anchors in DIFAIR led to a degradation of the AUROC score. This may come from the fact that in CAC, in addition to using distance to anchor during learning, there is another loss term to ensure that an instance's representation will be represented far from class anchors other than its own. In contrast, DIFAIR does not use such a loss term because of our specific objective of maintaining semantic meanings in representations. By avoiding the enforcement of instances being represented far from other anchors, DIFAIR allows for *semantic proximity* with other classes. This means that instance representations may exhibit the activation of certain features that are shared with other classes, promoting a more semantically meaningful representation. Therefore, the trade-off between forced separation and semantic proximity highlights the different objectives of these two methods, which can influence their respective AUROC score in OSR tasks.

# 5 REPRESENTATION ANALYSIS

## 5.1 VIZUALIZATION

To determine whether the learned representation extracts independent features on different dimensions of the latent space and turns off absent features, we generated diagrams from the representations in a similar way to Hinton diagrams (Hinton & Shallice, 1991). In this, this visualization, we inspected results obtained on the third split of CIFAR10. Figure 3a shows the diagram of the mean representation of correctly classified train instances from the cat class, both for a model trained with cross-entropy loss and for a model trained with DIFAIR. In the cross-entropy representation, it is not possible to directly associate a feature with a class. One way to achieve this would be to compute the mean activation for a class, then associate the most activated features with that class. Notably, some of the most activated features in the cat mean representation overlap with those of the dog class (as visible in Appendix E.2), which means that there are shared dimensions across classes. In the DIFAIR representation, the association between features and classes is clear. It is known which features should be activated for an instance to be classified into a particular class. Features are not shared between classes, but dog features are slightly activated in the cat mean representation, exhibiting some semantic similarity between the two classes.

The mean representation of DIFAIR shows that, for a given class, all features have the same activation. Additionally, it was observed that these features are consistently activated, even though not all features describing a class are always present in an image. Initially, the aim of optimizing using a hypersphere was to enable the model to learn the expression of different features per dimensions. However, obtained results suggest that the information is duplicated on dimensions. To verify this hypothesis, we examined the weights of the convolution layer before average pooling. We supposed that, if values on multiple dimensions in the representation are the same, weights used to compute those values (referred to as "class weights") must be the same (i.e. the same information is selected from the previous layer's output). To check whether class weights converge to the same value, we computed the mean standard deviation of those weights (Appendix E.1). Confirming our hypothesis, results showed that the standard deviation is around $0.015$ for each group of class weights, while it is around $0.15$ across the whole set of weights. Figure 3b shows a decrease in the standard deviation for each class weights, showing that they indeed converge during learning. Moreover, results from this graphic show that the learning rate restarts accentuate the convergence of these weights.

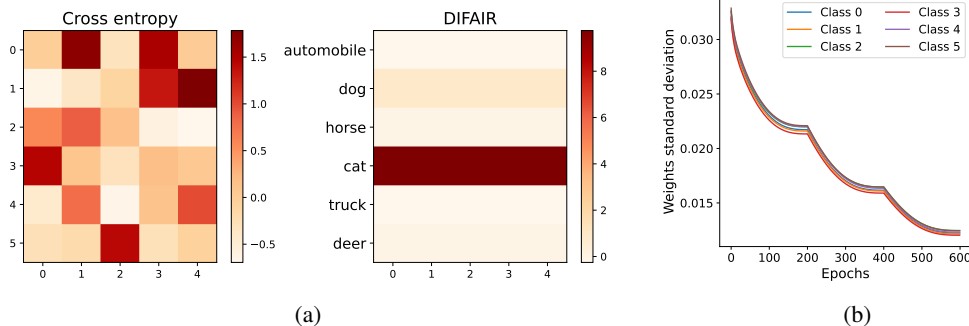

Figure 3: Results from models trained on the third CIFAR10 split. (a) **Comparison of the mean cat representations obtained using cross-entropy and DIFAIR.** DIFAIR features can be directly associated with classes, unlike cross-entropy features. (b) **Training evolution of the standard deviation per group of weights.** The standard deviation is used for assessing weight convergence.

## 5.2 AREAS FOR IMPROVEMENT

To avoid feature duplication, an attempt was made to add a loss term to maximize the standard deviation of each class weights (Appendix E). The weights and features obtained indeed have different values. However, features of the same class are still activated with close values. We expected that forcing the extraction of different features in this way would lead to features that are more or less activated depending on the characteristics present in the input image but that was not the case. Instead of intra-class feature duplication, it would be preferable to observe feature duplication over other classes. This would signify that there are shared features among the classes. The use of a hypersphere around the anchor allows other class features to be activated with a tolerance. If the activation of another class's feature is too high, it will place the instance outside its hypersphere. While the loss function allowed such feature activation, this behavior has not emerged from the optimization and we need to determine how to constrain it. Current investigations imply using a loss function that will be more tolerant than the Euclidean distance to the activation of other classes' features. Since MOS is an evaluation metric that is better suited than distance for OSR tasks, it inspired us to incorporate a term in the loss, using the magnitude along the dimensions associated with a class.

## 6 CONCLUSION

In this paper, we presented our propositions to obtain more interpretable representations using neural network classifiers, where each extracted feature is associated with a specific class, while a separation between different classes is maintained in the representation space. To reach such objectives, we proposed the DIFAIR approach which relies on the definition of *class anchors* in the representation space, around which respective class instances should be represented within the boundaries of a hypersphere. To optimize a neural network in this setting we proposed the usage of the Euclidean distance thresholded by the hypersphere radius.

The task of Open-Set Recognition is used to evaluate the quality of our learned representations. Obtained results show that our method achieves results equivalent to Class Anchor Clustering, a similar approach, when using the Maximum Output Score as OSR score. Using the distance to anchors to detect unknown classes was less efficient in our setup. Using the distance during training allows the activation of features from other classes, since the instance can be represented within a hypersphere, reducing class separation in the representation space to benefit from more semantically meaningful representations. The visual analysis of the learned representations demonstrated that extracted features are duplicated across class dimensions, while we aimed at obtaining distinct features.

While we did not fully realize our objectives, DIFAIR results unveiled multiple ways of improvement towards representation interpretability. Future investigations will focus on exploring loss functions tolerant to both the absence of features on true class dimensions, and the detection of the same feature in multiple classes.

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

# A IMPLEMENTATION DETAILS

## A.1 VGG32 ARCHITECTURE

For all experiments in this paper we used the neural network architecture proposed by Neal et al. (2018), which is commonly used for open-set recognition. It is referred to as VGG32 because it is a lightweight model similar to the VGG architecture (Simonyan & Zisserman, 2014). We implemented this architecture and all experiments using TensorFlow 2.12.1.

The base building block of this architecture is composed of a $3 \times 3$ convolution, followed by batch normalization and Leaky ReLU with a slope coefficient of 0.2. Each convolution is parameterized by a number of filters and a stride.

Three of those base building blocks are then stacked to create an "encoder" block. Dropout with a probability of 0.2 is added at the beginning of each encoder block (dropout is applied on the input). At the end of each encoder block, spatial dimensionality is reduced using a stride of 2 on the last convolution.

Finally, three "encoder" blocks are stacked to form the full architecture. There are $64$ convolution filters for the first two convolution layers and $128$ for the remaining layers, resulting in a model with 1 million parameters. Global average pooling (Lin et al., 2013) and a fully connected layer are added on top of this stack for the baselines model. In our approach, the fully connected layer is removed and a convolution layer with $\mathcal{N} \times$ number_of_classes filters is added right before global average pooling.

No bias are used in any layer, following Neal et al. (2018) proposition and standard uses of this architecture in OSR.

## A.2 OPEN-SET EXPERIMENTS

All experiments were run on a HPC cluster, on different NVIDIA graphics cards (Tesla V100 and Quadro RTX 6000). Training took a maximum of 2 hours per dataset split. We used the training protocol proposed by Vaze et al. (2022), for which they found the hyperparameters on a validation set, for a single split. The purpose of using this protocol was to have a strong baseline against which to compare and to train efficient networks, following the conclusion of the authors that closed-set accuracy is correlated to open-set performances. Hereafter we describe the complete protocol set up for our experiments.

**Data augmentation.** For every dataset but MNIST, we used RandAugment (Cubuk et al., 2020), a state of the art data augmentation method parameterized by the number of variations, $N$, applied to an image and the magnitude, $M$, of the variations. $N$ and $M$ values found by Vaze et al. (2022) are used. The TensorFlow implementation of RandAugment[1] that we used does not work on one channel images. Thus, we could not use it on MNIST and instead applied contrast and brightness augmentations, showing equivalent results. Afterwards, images are randomly cropped, and horizontally flipped when datasets do not contain numbers.

**Data preprocessing.** Images go through data augmentation in pixel format and come out in the same format. The first layer of our network rescales images to the $[0, 1]$ range, then a layer normalize data to have a mean of $0$ and a standard deviation of $1$. For normalization, we used a mean and variance computed on the training dataset, therefore they are different for each split.

**Additional details.** Contrarily to Vaze et al. (2022) we did not implement label smoothing (Szegedy et al., 2016) since it was only used when training on Tiny ImageNet and we could not reproduce their results on this dataset, even when they did not use label smoothing.

We noticed that two of the splits are the same for CIFAR10 but we did not change them in order to stick to the protocol proposed by Vaze et al. (2022).

---

[1]https://www.tensorflow.org/api_docs/python/tfm/vision/augment/RandAugment

**Method parameters.** Both for CAC and our method, the value on anchor dimensions is set to $\alpha = 10$. In DIFAIR, we allocated $\mathcal{N} = 5$ dimensions per class and used a hypersphere radius $r = 0.4 \times \sqrt{2\mathcal{N}\alpha^2} \approx 12.649$. DIFAIR hyperparameters were found following the protocol described in Appendix C.

**Training.** Following Vaze et al. (2022) protocol, we trained all models (baselines, CAC and DIFAIR) with a batch size of 128 for 600 epochs, with an initial learning rate of $0.1$ except for Tiny ImageNet where we used $0.01$. They proposed to use a cosine annealed learning rate with restarts (Loshchilov & Hutter, 2016) at epochs 200 and 400, and a learning rate warmup, linearly increasing the learning rate value from 0 to the initial value at epoch 10. Stochastic Gradient Descent (SGD) optimizer with momentum $\mu = 0.9$ and weight decay $\lambda = 1e - 4$ was used.

The images used are $32 \times 32$ pixels in size, except for Tiny ImageNet where they are $64 \times 64$ pixels. Models are trained on the full training split, no validation data is held out since hyperparameters are already chosen and we do not use early stopping.

## B  HYPERSPHERE TOLERANCE

In DIFAIR, the objective is to obtain representations that bear semantic meaning. For this reason, instead of forcing instances to be represented at an anchor point in the representation space, we enable the representation of instances within a hypersphere centered at the anchor. Using hyperspheres allows instances to be represented both close to their true anchor and also in the direction of some semantically similar class.

To better explain our intuitions, we choose a representation as an example and calculate its distance to anchors. We define a situation where there are 3 classes, and DIFAIR parameters are $\alpha = 10$, $\mathcal{N} = 2$ and $r = 8$. In this situation, class anchors are defined as in Eq. 1, reproduced below.

$$\mathcal{A}^1 = (\alpha, \alpha, 0, 0, 0, 0) \quad \mathcal{A}^2 = (0, 0, \alpha, \alpha, 0, 0) \quad \mathcal{A}^3 = (0, 0, 0, 0, \alpha, \alpha)$$

Let $z$ be the representation of an image containing a cat. The Euclidean distance between $z$ and class anchor $\mathcal{A}^1$ is:

$$d = \sqrt{(\alpha - z_1)^2 + (\alpha - z_2)^2 + z_3^2 + z_4^2 + z_5^2 + z_6^2}$$

In this situation, the distance is going to be 0 if $z = \mathcal{A}^1$. Now, say that one dimension not associated with the first class is activated, for example, $z = (\alpha, \alpha, 4, 0, 0, 0)$. The distance to the anchor is 4, so this representation is within the allocated hypersphere. $z_3$ can go up to $8$ if other values do not change while remaining within the hypersphere of $\mathcal{A}^1$.

If $\mathcal{A}^1$ was the cat class anchor, and $\mathcal{A}^2$ the dog class anchor, $z$ would be both close to its anchor, within the hypersphere, and closer from the dog class than from the third class. Our $z$ representation is valid while bearing semantic information because a dog feature was activated when detecting a cat, representing a shared feature. Even though it means duplicating the information compared to distributed representations (Hinton, 1984), this has the advantage of allowing an interpretation because features are class-associated. In this case, it shows that there is a feature common to both classes.

Therefore, changing the hypersphere radius $r$ influence how *tolerant* DIFAIR is to the activation of other class features, and influence how much semantic is allowed in the representations.

## C  HYPERPARAMETER SEARCH

Our method uses three hyperparameters. Two of them describe the anchors: $\mathcal{N}$ is the number of dimensions allocated per class, $\alpha$ is the target value on dimensions associated with a representation. Lastly, $r$ is the radius of hyperspheres, centered at each anchors, within which predictions are accepted during training. A hyperparameter search was conducted to investigate the influence of those parameters on closed-set performances.

We searched hyperparameters giving the best results in the closed-set setting because the main objective of DIFAIR is to have a method capable of extracting independent features, representative of

each class from the training data. We hypothesized that the extraction of such features would be efficient for open-set recognition but it is not a direct optimization objective. For this reason, we did not optimize hyperparameters for open-set performance.

Since we use the Euclidean distance for optimization, all anchors are separated by a distance of $\sqrt{2\mathcal{N}\alpha^2}$ because there are $2\mathcal{N}$ non-null values on different dimensions in two anchors. Following the same reasoning, the distance from an anchor to the origin of the representation space is $\sqrt{\mathcal{N}\alpha^2}$.

In our approach, we aim to prevent the allocated hyperspheres from overlapping. This would leave no space for unknown data to be represented between two anchors and the separation between classes would be difficult to achieve. Therefore, the upper bound for $r$ is $0.5 \times \sqrt{2\mathcal{N}\alpha^2}$. The combinatorial search carried out showed that $r$ values greater than half the distance between anchors do not allow the model to converge.

**Selection criteria.** To evaluate the quality of the representation learned without evaluating OSR tasks, we report the closed-set accuracy and a metric evaluating the *hypersphere occupancy*. This metric is the percentage of representations that are located in their associated hypersphere. It is used to evaluate to which extent the optimization validated our loss criteria. The loss value could not be used directly because it is not normalized across experiments. This is because the distance between elements in the representation space varies depending on the hyperparameters.

**Experiment details.** We chose to fix $\mathcal{N} = 5$, hypothesizing that the more features are added, the better the class will be described. This is a logical conclusion if the premises that extracted features are independent and representative of a class are respected. An object will be better described with more features. There certainly is a limit on the number of features that can be extracted for a class, but we have not studied such extreme cases yet.

All models were trained on the CIFAR10 train dataset (using all classes), with 20% of the training data used as a validation set, on which results are reported. No data from the testing set was used. All experiments were run only one time due to the quantity of models that had to be trained.

**Discussion.** First, we studied the implications of variations of $\alpha$ and $r$ for a range of values, results can be seen in Figure 4a and 4b. The results at the top-left corner of each diagram confirm that having a hypersphere radius that is greater than half the distance between anchors is not suitable for optimization. The reason is that hyperspheres overlap. Instances are more likely to be within their hypersphere from the start, causing no optimization and low accuracy. The best results obtained, excluding the cases where hyperspheres overlap, are on the diagonal, when the radius is close to $\frac{1}{3}$ of the distance between anchors. Under $\frac{1}{3}$, the accuracies and hypersphere occupancies are decreasing. Finally, augmenting the value on dimensions, $\alpha$, seems to increase the accuracy on the diagonal.

In a second experiment, we selected the 3 largest $\alpha$ values from the first experiment and investigated further the value of $r$, making it range from $\frac{1}{3}$ of the distance between anchors to $\frac{1}{2}$ this distance. Obtained results are visible in Figure 4c and 4d. In terms of hypersphere occupancy, the closer the radius coefficient is to $0.5$, the better the results. But in our approach, we want to have some space between hyperspheres where unknown data can be represented. In terms of accuracy, results are slightly improved by increasing the coefficient from $\frac{1}{3}$ of the distance between anchors. It follows that we do not want the radius coefficient to be too close from $\frac{1}{3}$ or $\frac{1}{2}$, thus we selected the value $0.4$. For this value, results on the row are close for $\alpha = 8$ and $\alpha = 10$. Keeping in mind that these are just one-run results, both values could have been selected but we chose $\alpha = 10$. That is the same value that was selected for $\alpha$ in CAC (Miller et al., 2021).

Based on our hypothesis on $\mathcal{N}$ and these experiments, we chose $\mathcal{N} = 5$, $r = 0.4 \times \sqrt{2\mathcal{N}\alpha^2}$ and $\alpha = 10$. Those values were used for all conducted experiments in this paper.

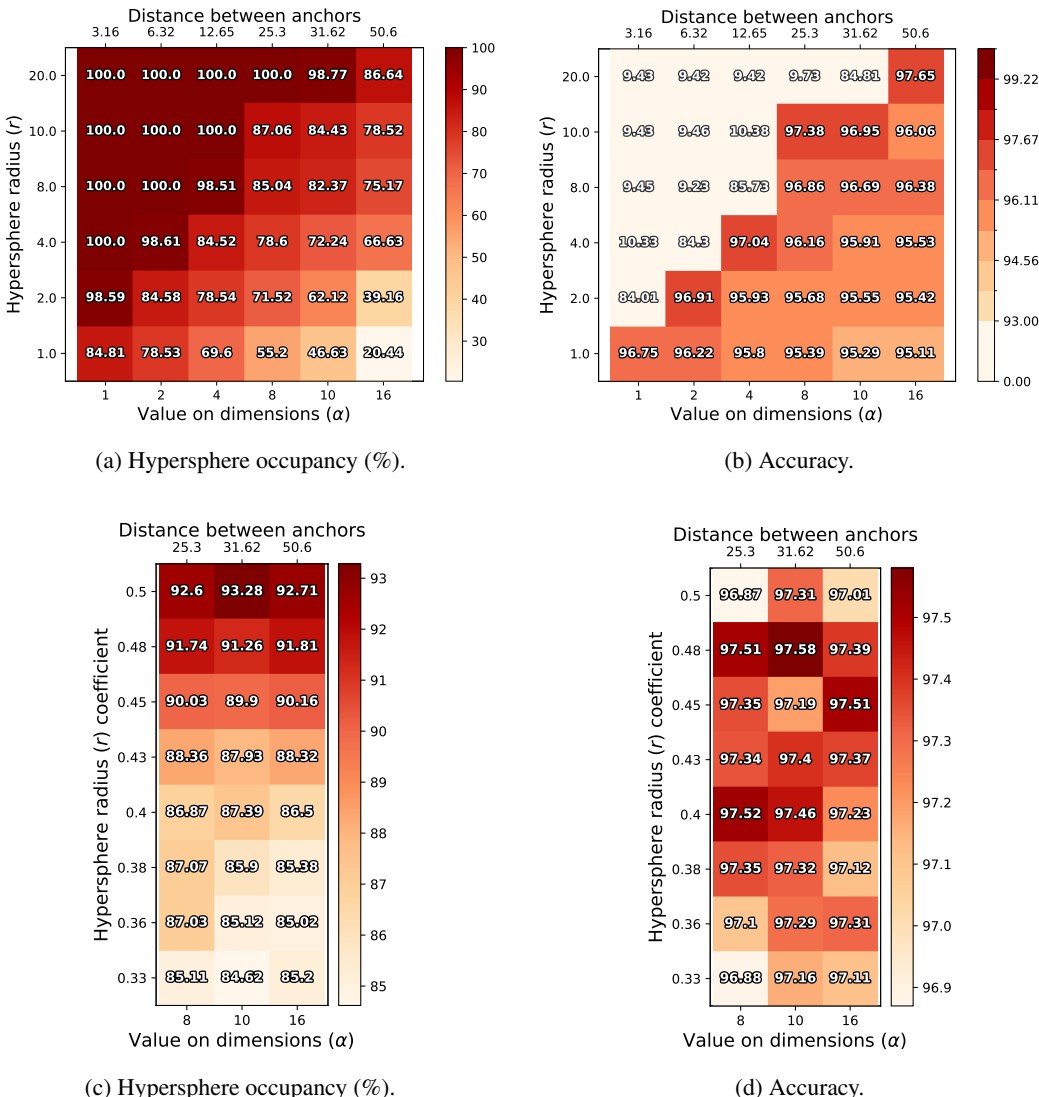

(a) Hypersphere occupancy (%).

(b) Accuracy.

(c) Hypersphere occupancy (%).

(d) Accuracy.

Figure 4: **Results obtained in closed-set during the hyperparameter search.** (a) and (b) show the results of the first combinatorial search; (c) and (d) show the results of the detailed search after a first elimination of possible hyperparameters.

# D    CLOSED-SET RESULTS

We report in Table 2 the closed-set performances of the models trained in section 4. It is interesting to note that while all approaches reached similar accuracies (except on TinyImageNet), they also had very different AUROC scores on the OSR benchmark (OSR results are reported in Table 1).

The cross-entropy loss-trained models was used to produce OSR results for MSP, MLS, and CELR.

Table 2: **Closed-set accuracy on Neal et al. (2018) benchmark**. We report the average closed-set accuracy (%) and the standard deviation, over the five known/unknown class splits.

| Method | MNIST | SVHN | CIFAR10 | CIFAR+10 | CIFAR+50 | TinyImageNet |
|---|---|---|---|---|---|---|
| Cross-entropy | 99.8±0.1 | 97.7±0.2 | 96.1±1.2 | 96.1±0.2 | 96.1±0.1 | 61.4±3.1 |
| CAC | 99.8±0.1 | 98.2±0.1 | 96.1±1.0 | 96.2±0.3 | 96.3±0.1 | 66.1±2.3 |
| DIFAIR | 99.8±0.1 | 98.0±0.2 | 95.8±1.2 | 96.2±0.3 | 96.2±0.1 | 66.8±2.3 |

# E    MORE VISUALIZATION OF REPRESENTATIONS

In this section, we present some supplementary results obtained with DIFAIR. We also compare them to the preliminary results obtained by adding a loss term on weights of the last convolution layer, in order to prevent weights convergence and thus prevent the extraction of the same features for a class representation. To clarify discussions, we refer to the method presented in this article as DIFAIR, and to the same method with the additional loss term as DIFAIR-std.

## E.1    LOSS ON WEIGHTS

Since we observed the convergence of groups of weights on the last convolutional layer using the standard deviation, the straightforward way to avoid this convergence is to add a loss term depending on the standard deviation. We need a function that decreases as its input, the standard deviation, increases. The function proposed is described in Eq. 4. Our approach extracts $\mathcal{N}$ features per class. Let $s$ be the number of weights used to compute one feature of the representation. In our case, $s = 3 \times 3 \times 128$ because on the last convolutional layer, filters are of dimensions $3 \times 3$, and the penultimate convolutional layer had 128 filters. The weights used to compute *one feature* are flattened in the following calculations. Let $\boldsymbol{W}^c \in \mathcal{M}_{s \times \mathcal{N}}(\mathbb{R})$ be the set of weights responsible for the extracted features of the $c^{\text{th}}$ class. The loss on weights $\mathcal{L}_{\text{weights}}$ is computed as follows:

$$\mathcal{L}_{\boldsymbol{W}^c} = 2 \times \exp\left(-3 \times \frac{1}{s}\sum_{i=0}^{s-1}\text{std}(\boldsymbol{W}_{:,i}^c)\right) \tag{3}$$

$$\mathcal{L}_{\text{weights}} = \sum_{c}\mathcal{L}_{\boldsymbol{W}^c} \tag{4}$$

Where $\text{std}$ is the function computing the standard deviation. The coefficients 2 and -3 where chosen so that $\mathcal{L}_{\boldsymbol{W}^c}$ decreases and reaches values close to 0 when $\frac{1}{s}\sum_{i=0}^{s-1}\text{std}(\boldsymbol{W}_{:,i}^c)$ approaches 1.5. We remind that these are only preliminary results, further investigations are required.

## E.2    REPRESENTATIONS OF DIFFERENT MODELS

As in Section 5, we study the third split of CIFAR10, in which known classes are: automobile, dog, horse, cat, truck, and deer. Unknown classes are: *ship*, *bird*, *airplane*, and *frog*. Here, we present the mean representations of all classes for a model trained with DIFAIR and for a model trained with cross-entropy loss. Mean representations are calculated from the representations of correctly classified instances.

Figure 5a illustrates the mean representations learned by DIFAIR for each class. It shows that, on average, the cat and dog representations have common features that are activated. The average activation value for cat features and dog features in their respective representations is lower than for

the other classes. We presume that the model has difficulty in extracting features with certainty for these classes but further investigation is required.

In the DIFAIR approach, extracted features are directly associated with classes. We explained in Section 5 that in cross-entropy learned representations, the most activated features of a class mean representation could be associated with this class, but it is not a trivial task since features can be activated for multiple classes. The mean class representations obtained from a model trained with cross-entropy loss are shown in Figure 5b. It can be noticed that some features are activated both in cat and dog mean representations.

Comparing DIFAIR and DIFAIR-std performances, DIFAIR obtained an accuracy of 93.5% and AU-ROC scores of 77% using distance to anchors score, 85% using the Maximum Output Score (MOS). The results obtained with DIFAIR-std are an accuracy of 93.6% and AUROC scores of 72.9% using distance to anchors score, 84% using MOS. Further tests are required, but an initial comparison shows that DIFAIR-std performs similarly to DIFAIR, except when the AUROC is measured using the distance to the nearest anchor. Figure 5c shows a representation obtained with DIFAIR-std for an image containing a frog, which is an *unknown* class. The features extracted in this representation have slightly different values, which shows that the loss term has avoided convergence of the weights. Most importantly, this figure shows that for an unknown class, all the dimensions of two classes are activated. This implies that the attributes represented on the dimensions of a class are not independent. This is a behavior we do not want to have in our approach and will require further work. Mean class activations are not reported for DIFAIR-std as they are similar to those of DIFAIR, the only difference being slight variations across class-associated features whereas for DIFAIR all values are equal.

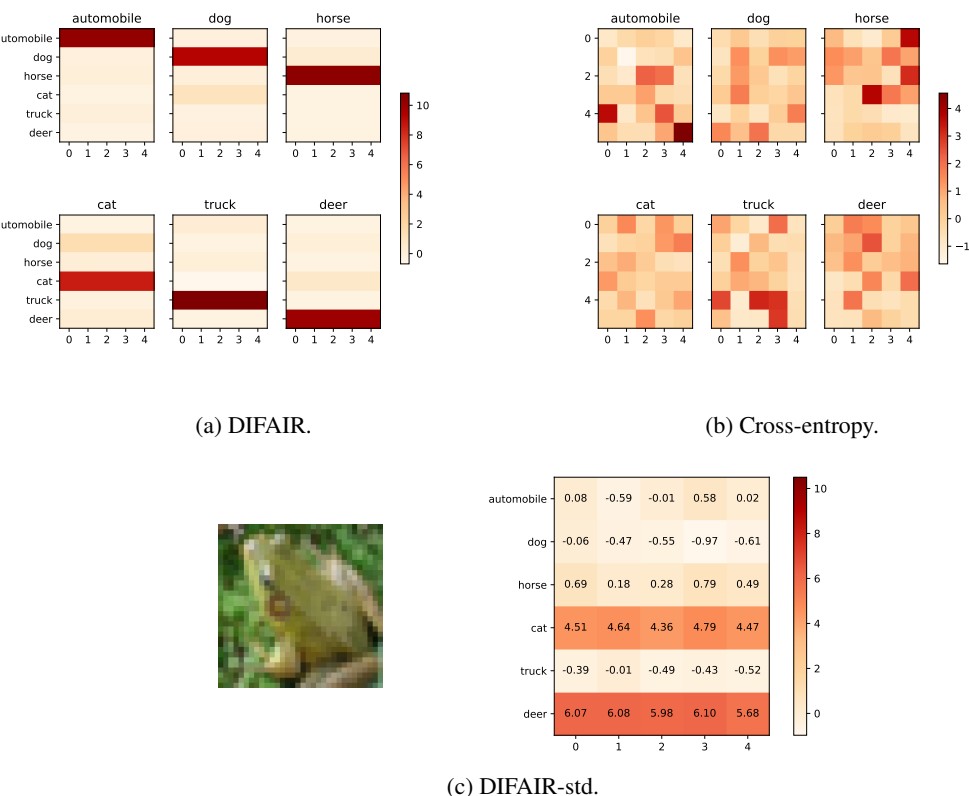

(a) DIFAIR.

(b) Cross-entropy.

(c) DIFAIR-std.

Figure 5: **Representations for different models.** (a) Mean representations per class of a model trained with DIFAIR. (b) Mean representations per class of a model trained with cross-entropy loss. (c) Example of a representation of the DIFAIR-std model given an image containing a frog, an unknown class.

### E.3 T-SNE VISUALIZATION

In addition to visualising average representations, we observed the distribution of instances in the representation space using t-SNE, a dimension reduction method. We compare the distribution of *known* class representations between DIFAIR and DIFAIR-std in Figure 6a and 6b respectively. It seems that in both cases the distinction between animals and vehicles is clear in the representation space. Furthermore, there is no obvious distinction between the two approaches. In these figures, the anchors and the average representations are also shown in order to situate the organisation of the instances around the anchors.

We also compare the distribution of *unknown* classes in the representation space for both methods in Figure 6c and 6d. The aim of this visualization is to check whether it is possible to group together the unknown classes in the representation space, on the basis of the attributes extracted by our model. Many instances are represented around the anchors, which explains the low performance in OSR when distance is used as a score. The distance between these instances and the anchors being too low, it leads to a prediction. However, we find that there is a non-random distribution of unknown classes. Ship instances are generally represented closer to known vehicle' anchors, as are airplane instances, but they also appear to be represented around animal anchors. Unknown animal classes also appear most often on the animal anchors' side. Again, there is no noticeable difference between the results obtained by DIFAIR and those obtained by DIFAIR-std.

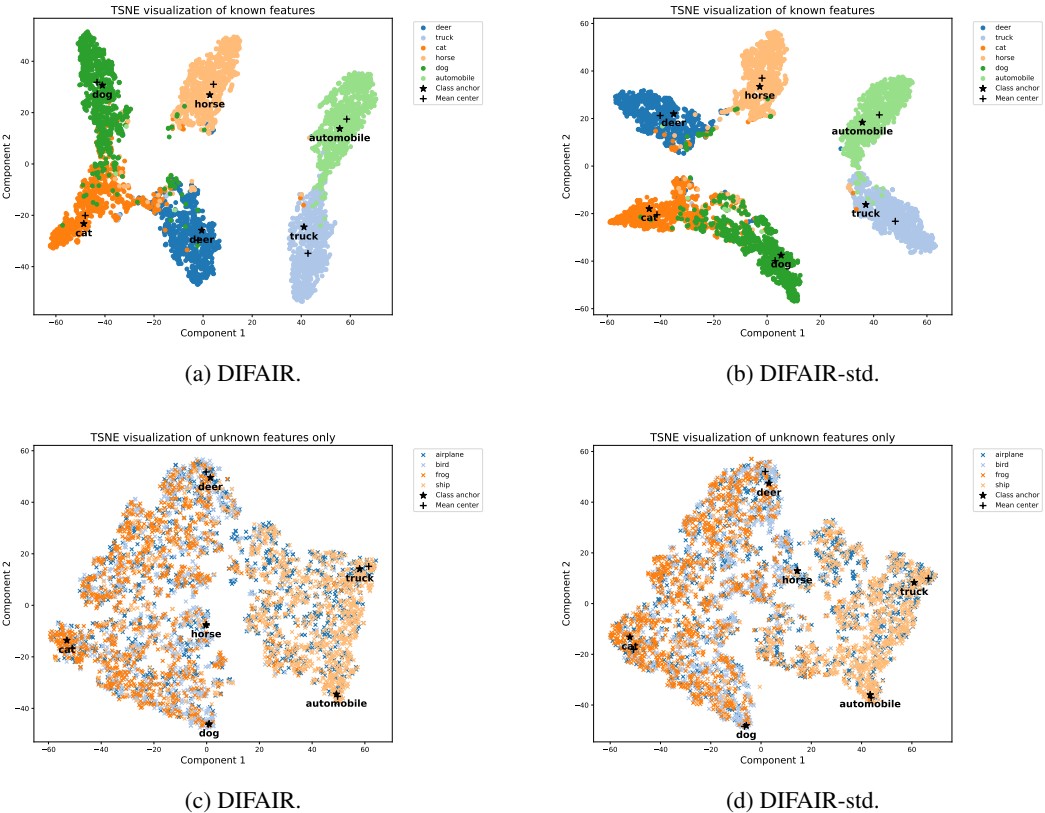

Figure 6: **t-SNE visualizations of representation spaces for DIFAIR and DIFAIR-std.** (a) and (b) show the distribution of known class in space. (c) and (d) show the distribution of unknown classes in space.

