# OpenReview forum: "DIFAIR: Towards learning differenciated and interpretable representations"
_ICLR.cc/2024/Conference — Submitted to ICLR 2024_

### Official Review · Reviewer_FwaY · 2023-10-31

**Soundness:** 3 good
**Presentation:** 2 fair
**Contribution:** 2 fair
**Rating:** 5
**Confidence:** 3

**Summary:**

In this paper, the authors proposed a method called DIFAIR for learning differentiated and interpretable representations. The proposed method involves: (1) modifying a given convolutional neural network by removing the classification head and appending a convolutional layer with N*K filters (N = the number of feature dimensions allocated for each class, K = the number of training classes) followed by a global average pooling, which results in a feature vector z of dimension N*K; (2) representing each class with a radius r hypersphere around an N*K-dimensional class anchor vector, which is 0 everywhere except the the dimensions assigned to that class; (3) training the network by minimizing the Euclidean distance between the feature vector z and the class anchor, if z lies outside the hypersphere of radius r centered at the anchor. The authors applied DIFAIR to open-set recognition (OSR) benchmarks used by Neal et al. (2018), and compared their approach with competing state-of-the-art OSR methods.

**Strengths:**

- The proposed DIFAIR is technically sound, and does provide promising results on OSR problems.
- The proposed method requires no additional data for "known unknowns," unlike DCHS and (ARPL+CS)+.

**Weaknesses:**

- Even though the authors claimed that their approach would yield "interpretable" representations, the semantic meaning of each feature dimension in the representation space is unclear. I would recommend scaling back on the claim with regard to "interpretability."
- While the proposed DIFAIR is promising and requires no "known unknowns" during training, its OSR performance is not as good as simple competing methods such as MLS (especially on CIFAR10 and CIFAR+N).

**Questions:**

- It is not clear to me how you applied Maximum Output Score (MOS) on DIFAIR. Can you explain that? I would recommend including this information in the main paper.

**Details Of Ethics Concerns:**

N/A.

---

> ### Author Response · Authors · 2023-11-21
>
> Thank you for your time and feedback.
>
> > Even though the authors claimed that their approach would yield "interpretable" representations, the semantic meaning of each feature dimension in the representation space is unclear. I would recommend scaling back on the claim with regard to "interpretability."
>
> You are correct, our representations are not entirely interpretable, which is why we opted to temper our findings  by using ``towards’’ in the title. We will try to find a more appropriate word. However, our representation does provide some understanding of the prediction made and we can gain insights into what is incorrect in the network behavior. For instance, when a classification is incorrect, we can determine which class features are activated. This provides insights that the network classified incorrectly due to the activation of distinct features of other classes. Of course, the next step is to determine what represent the features.  However, it still remains challenging to explain why such features were detected. While this “why” question still holds, attempting to answer it was not the purpose of our approach.
>
> > It is not clear to me how you applied Maximum Output Score (MOS) on DIFAIR. Can you explain that?
>
> The Maximum Output Score, follows the same concept as the Maximum Logit Score. A different name was used as DIFAIR does not output logits but rather an image representation. The maximum value of the representation is selected and used as the OSR score. A high score would indicated the detection of a known class, with a highly activated attribute, while a low score signifies that no known class was detected.

---

### Official Review · Reviewer_8kv1 · 2023-11-02

**Soundness:** 1 poor
**Presentation:** 2 fair
**Contribution:** 1 poor
**Rating:** 3
**Confidence:** 4

**Summary:**

In standard deep image classification models (e.g. ResNet50, ViT), there is usually a linear layer producing class scores (usually 1 score per class).
This paper proposes to output N scores per class, instead of 1.
The network is then trained with $\ell_2$-distance over scaled N-hot labels (here I abuse the one-hot vector notation by saying N-hot as there are N class scores to regress now).
This way, sparse set of features are learned for each class, which, in the end, can better suit open-set recognition tasks.
However, unfortunately, all the results achieved by the proposed method are worse than the best compared baseline.

**Strengths:**

Learning interpretable representations is an important research problem, especially because more and more ML-based products are interacting with humans in daily life.
The paper is making an attempt to learn such representations, and it overall reads fine.
Open-set recognition experiments are performed on 6 datasets including MNIST, SVHN, CIFAR10, CIFAR+10, CIFAR+50 and TinyImagenet.

**Weaknesses:**

It seems that what is formulated as "fixed anchors" ($\mathcal{A}$) is actually scaled one-hot labels in supervised learning.
So the difference between the conventional $\ell_2$ regression of labels is just that the proposed method is regressing N of these scalars per class. I'm not sure if this is any interesting for ICLR audience.

Also, no performance improvement is observed, I'm not sure if the proposed offers any practical advantages.

Also, I think the paper is missing large bodies of related works.
- interpretability works, which try to disentengle representations (dimension) or associate certain dimensions to certain classes
    - Interpretable Explanations of Black Boxes by Meaningful Perturbation, Fong et al. 2017, ICCV, and many follow-ups
- metric learning works, which learn multiple linear sub-spaces for each class
    - SoftTriple Loss: Deep Metric Learning Without Triplet Sampling, Qian et al. 2017, ICCV, and many follow-ups.

Some other comments:
- Figure-2 does not offer much (if any). Why would the subplot on the left be a "standard" learning setting. If the class weights are orthogonal, it becomes what is depicted on the right. It would be nice to use this space with a figure showing the method diagram.
- Not sure if Section-3.3 offers much (if any). It would be nice to provide more experimental results, given that the current ones in Table-1 are already discouraging. Analysis on interpretability of representations could be an option.

**Questions:**

I would like the authors to address the weaknesses I mentioned above.
Regardless, I'm not sure if ICLR is a good fit for this submission.

---

> ### Author Response · Authors · 2023-11-21
>
> Thank you for your time and feedback.
>
> Although we did not manage to reach the same performances as other approaches, DIFAIR provide some interpretability via the association of representation dimensions with specific classes. By looking at the representations learned with DIFAIR, it gave us insight into the flaws of our approach, which is hardly possible when looking at a cross-entropy learned representation for example.
>
> As you and other reviewers have pointed out, further investigation into the interpretability of representations is necessary. Mainly, we will provide results on datasets other than CIFAR10, with larger representations and more classes.
>
> Thank you for providing those references. We will examine those fields and papers more closely.

---

### Official Review · Reviewer_QmgY · 2023-11-03

**Soundness:** 1 poor
**Presentation:** 1 poor
**Contribution:** 2 fair
**Rating:** 3
**Confidence:** 3

**Summary:**

This paper presents an idea of disentangling features of different classes to achieve open set recognition. A training method called DIFAIR is proposed to achieve this goal. Empirical validation on DIFAIR are provided.

**Strengths:**

The idea of disentangling features of different classes is reasonable.

**Weaknesses:**

1. It seems that the proposed method is only enforcing data representation to be within the spherical area around anchors of each class. Similar ideas have been explored in previous works such as [1] on other topics. Therefore, the author have to justify the extra insights provided by their work.
2. Empirical validation of the paper can be improved.
3. The writing of the paper can be improved. For example, in Figure 3 (a), it is not quite clear what are the x- and y- axis, and what does the value of the heatmap represents.
4. According to the Table 1 in the paper, it seems that the performance of the proposed method is far worse than the current state-of-the-arts.

[1] Targeted supervised contrastive learning for long-tailed recognition. T. Li, et al. Proceedings of the IEEE/CVF Conference on Computer Vision and Pattern Recognition.

**Questions:**

1. As far as I am concerned, DIFAIR only introduces an optimization objective to enforce representations being compact around anchors of each class. How can the method enforce the representations of unseen classes to be out of the spherical areas of the known classes?

---

> ### Author Response · Authors · 2023-11-21
>
> Thank you for your time and feedback.
>
>
> > 1. It seems that the proposed method is only enforcing data representation to be within the spherical area around anchors of each class. Similar ideas have been explored in previous works such as [1] on other topics. Therefore, the author have to justify the extra insights provided by their work.
>
> Regarding point 1, as far as I understand it at the moment, the approach in reference [1] uniformly distribute the representation of classes on a hypersphere centered at the origin of the representation space. Additionally, instances also appear to be represented on this hypersphere and must maintain proximity to the defined centers.
>
> In DIFAIR, the defined anchors are located on a hypersphere centered at the origin of the representation space, but they are not uniformly distributed on the hypersphere. These anchors are targets for the representations, where each dimension represents extracted features. The activation of the values on dimensions depend on the presence of a features in the input image. In the DIFAIR approach, the allocation of a hypersphere around the anchor allows attributes from other classes to be activated to maintain a semantic meaning in the representation. Furthermore, our approach provides more interpretable representations by associating each dimension of the representation with a class.
>
> > As far as I am concerned, DIFAIR only introduces an optimization objective to enforce representations being compact around anchors of each class. How can the method enforce the representations of unseen classes to be out of the spherical areas of the known classes?
>
> The proposed method does not directly enforce the representation of unseen classes to be outside the hyperspheres of known classes. The approach is based on the hypothesis that the values expressed on each dimension of the representation will represent the presence or absence of a feature in the image. This is due to the fact that when features of other classes are activated, the instance will be represented too far from its anchor. Therefore, if an image contains a known class, a maximum of features of that class should be detected and no features of other classes. This will ensure that the representation of the image will be placed near the corresponding class anchor (in the hypersphere). However, in the situation where an image contains an unknown class, no known features (learned on known data) should be detected, therefore placing the image out of allocated areas.
>
>
> [1] Targeted supervised contrastive learning for long-tailed recognition. T. Li, et al. Proceedings of the IEEE/CVF Conference on Computer Vision and Pattern Recognition.

---

> > ### Comment · Reviewer_QmgY · 2023-11-23
> >
> > Thanks for your reply. These partially resolve my concern, and I believe there are some novelty in this work. However, at this point, the presentation is still not mature enough to be accepted. Therefore, I will keep my score.

---

### Official Review · Reviewer_agmg · 2023-11-04

**Soundness:** 2 fair
**Presentation:** 4 excellent
**Contribution:** 2 fair
**Rating:** 3
**Confidence:** 4

**Summary:**

This paper introduces DIFAIR (Differentiated And Interpretable Representations), an approach designed to differentiate between known and unknown classes in Open-Set Recognition (OSR) and improve the interpretability of neural network classifiers. DIFAIR introduces class anchors in the representation space and optimizes the model to produce representations close to these anchors. Each dimension in the representation is associated with a specific class. The technique's performance is comparable to existing methods in OSR. Some simple visualization results are also provided to somehow show learned representation of DIFAIR extracts independent features on different dimensions.

**Strengths:**

1. The paper is generally well-structured and lucid in its exposition.

2. The concept of training a model where each dimension signifies specific visual features from certain classes is intriguing. This could not only lead to a model with strong OSR performance but also pave the way for more transparent and interpretable deep visual models.

**Weaknesses:**

1.   Closed-set classification performance.

DIFAIR's fundamental premise is that each representation dimension corresponds to a specific class. To accomplish this, DIFAIR trains an anchor for each class, ensuring that only certain dimensions are activated for that class. While this might enhance Open-Set Recognition (OSR) and model interpretability, it raises the question of whether such a model could maintain the strong closed-set classification performance of conventional deep visual models, particularly when trained on large-scale datasets like ImageNet. The authors suggest that the hypersphere around each anchor provides some tolerance for activating other class features, but there seems to be a trade-off between interpretability (requiring low tolerance) and closed-set classification performance (requiring high tolerance). It's unclear if DIFAIR can easily find a balance point that provides both satisfactory closed-set classification performance and good interpretability.

2. Lack of theoretical analysis.

The paper lacks essential theoretical support and analysis. Many ideas seem to spring from intuition. While hypotheses and assumptions are discussed in Section 3.3, and some are empirically verified, none of them is well theoretically supported.

3. Experiment and visualization limitations.

3a. The evaluation of OSR performance is solely based on one model, VGG32. The paper's claims would be more persuasive if additional models, including CNNs and ViTs, were evaluated.

3b. The OSR performance of DIFAIR, as reported in Table 1, does not demonstrate significant improvement over previous OSR methods.

3c. Justification for the improved interpretability of DIFAIR is only provided through simple visualizations based on CIFAR10. More convincing evidence, such as visualization results from a dataset with more classes (like TinyImageNet), visualizations showing shared visual features in multiple classes and so on, would be beneficial.

In summary, the lack of theoretical analysis and the absence of compelling experimental and visualization results, lead me to lean towards rejecting the paper at this moment.

However, I am open to further discussions and potential rebuttals from the authors that may address these concerns.

**Questions:**

See weaknesses.

**Details Of Ethics Concerns:**

I do not find any particular ethics concerns.

---

> ### Author Response · Authors · 2023-11-21
>
> Thank you for your time and feedback.
>
> 1.  To be precise, we do not train an anchor (as other approaches like DCHS [1] do), but we fix it in advance so as to bias the training process toward building representation within a certain space around the pre-defined anchors. In Table 1, we refer to Appendix D, where we show that the closed-set performance of DIFAIR is similar to the closed-set performance of a model trained with cross-entropy. But you are right in saying that it should be tested on larger scale datasets to see if the trend holds. \
> Regarding the tolerance provided by the hypersphere, I would rather say that there is a trade-off between semantic in the representations and open-set classification. To have more semantic meaning, some attributes must be shared between classes. The radius of the hypersphere defines how tolerant the loss is to the activation of other classes features. However, if the tolerance is high (larger radius), class instances can be represented far from anchors, which is the behavior excepted for unknown instances. This can result in a lower open-set performance.
>
> 2.  It seemed hard for us to theoretically support our hypothesis, we developed intuitions on widely accepted hypothesis on neural networks like the extraction of features by CNNs. We will try to find more references for the basis of our reflections. Do you have any ideas or references in mind that provides theoretical support on subjects such as interpretability or semantic in representations?
>
> [1] From anomaly detection to open set recognition: Bridging the gap, Cevikalp et al., 2023

---

> > ### Comment · Reviewer_agmg · 2023-11-22
> >
> > After reviewing the authors' rebuttal, my initial concerns from my original review persist. Thus, I will keep the score.

---

### Official Review · Reviewer_xU1y · 2023-11-05

**Soundness:** 3 good
**Presentation:** 2 fair
**Contribution:** 2 fair
**Rating:** 3
**Confidence:** 4

**Summary:**

This paper proposes a new loss term to address the open-set recognition task. The loss term enables a more interpretable representation and the correspondence different dimensions of the learned feature and the classes can be interpreted. Experiments on the multiple datasets verify the effectiveness of the proposed method.

**Strengths:**

- The proposed loss term is well-motivated and technically sound.

- Experiments on multiple datasets verify the effectiveness of the proposed method and analysis has been conducted to study of the designs.

- The paper is overall well presented.

**Weaknesses:**

- The technical contribution is limited. The main contribution of this paper is a loss term, which however has not been extensively proved effective for a broad range of applications and for various methods.

- The evaluation is not comprehensive. Experiments only on small small-scale datasets are conducted, making the evaluation less convincing. Results on large-scale datasets should be conducted and more extensive analysis should be performed to have a more comprehensive evaluation.

-  The proposed method falls behind with many of the existing methods as shown in Table 1.

**Questions:**

See the weakness above.

---

> ### Author Response · Authors · 2023-11-21
>
> Thank you for your time and feedback.
>
> You are correct in stating that the loss term should be extensively tested on other applications and datasets. However, our goal was to present our first attempt to accomplish the objectives we set. Specifically, to train the network to obtain a more interpretable representation by associating dimensions with classes, while also attempting to retain some semantic meaning.
>
> We believed that the proposed loss term (Euclidean distance with a threshold), which seemed to be the simplest approach to distribute instances within hyperspheres around anchors in the representation space, could achieve these goals. The tests carried out on the OSR task and the representation visualizations allowed us to identify drawbacks of our approach. Therefore supporting the importance of being able to extract information from the representation.
>
> In this situation, we were not yet interested in testing on a larger scale, but we preferred to clearly state our hypotheses and present what needs to be addressed in future work to achieve our goals.
>
> In your comment :
>
>  > Results on large-scale datasets should be conducted and more extensive analysis should be performed to have a more comprehensive evaluation.
>
> I did not quite understand what you meant by ``more extensive analysis’’. Did you mean an analysis of the representations? Or were you referring to testing on other applications and datasets?

---

### Official Review · Reviewer_Veq2 · 2023-11-22

**Soundness:** 2 fair
**Presentation:** 3 good
**Contribution:** 2 fair
**Rating:** 5
**Confidence:** 3

**Summary:**

In this paper, the authors focus on learning such that results in interpretable representation, that has class-associated features and is free from distributed representations.  To that end the authors introduce a loss function targeting an optimization of the network’s learned representation, aligning it with the constraints specified for the representation space by defining an association between specific dimensions and use OSR tasks to assess the quality of the learned representation. They evaluate the proposed approach on open-set classification problem and provide insights into the model’s behavior compared to representations derived from standard mode of learning.

**Strengths:**

-The exposition of the paper is well done.

-The paper seems well motivated.

-The proposed loss seems reasonable in the context of interpretability for sematic reasoning and discrimination.

**Weaknesses:**

-The proposed approach seems like a build up on the OSR approach.

-Besides open set classification, and the design study of the proposed approach I was not able to see other experimental results (I see the closed-set results in the Appendix)

-In Table 1 it seems that the presented results for DIFAIR, does not demonstrate significant improvement compared to the other methods.

-I would have expected that in the empirical evaluation that the proposed approach is compared with similar approaches, e.g., other Self-Supervised Learning methods or their equivalent supervised counterparts.

**Questions:**

How does the method differentiates and compares with other Self-Supervised Learning Methods or their equivalent Supervised counterparts?

---

### Author Response · Authors · 2023-11-21
**Global response to reviewers**

We appreciate the reviewers’ time and comments on our article. As several reviewers noted, we plan to revise our work to include additional experimental validation of our hypothesis, test across other datasets and architectures, and provide theoretical support where applicable.

Though our results are not at the state-of-the-art level, we believe that our ideas are promising. This is supported by the fact that we managed to identify several areas for improvement while analyzing the representations learned using DIFAIR.

---

### Meta-Review · Area_Chair_tJiC · 2023-12-15

**Metareview:**

The paper addresses the problem of novel classes by proposing properties that should a representation have to model novel classes in an interpretable manner. Six reviewers provide details on what the paper is lacking in all aspects including the method design, empirical investigation, and importantly the presentation. The AC recommends rejection and urges the authors to attend to all the valuable comments for the next submission.

**Justification For Why Not Higher Score:**

The paper is not ready for publication for various reasons as outlined by the six knowledgeable reviewers.

**Justification For Why Not Lower Score:**

N/A

---

### Decision · Program_Chairs · 2024-01-16

Reject